# The Differences of Mechanisms in Antihypertensive and Anti-Obesity Effects of Eucommia Leaf Extract between Rodents and Humans

**DOI:** 10.3390/molecules28041964

**Published:** 2023-02-18

**Authors:** Sansei Nishibe, Hirotaka Oikawa, Kumiko Mitsui-Saitoh, Junichi Sakai, Wenping Zhang, Takahiko Fujikawa

**Affiliations:** 1Faculy of Pharmaceutical Sciences, Health Sciences University of Hokkaido, Ishikari 061-0293, Hokkaido, Japan; 2Faculty of Pharmaceutical Sciences, Suzuka University of Medical Science, 3500-3 Minamitamagaki-cho, Suzuka 513-8670, Mie, Japan; 3Faculty of Health and Sports, Nagoya Gakuin Unversity, 1350 Kamishinano, Seto 480-1298, Aichi, Japan; 4Faculty of Acupuncture & Moxibustion, Suzuka University of Medical Science, 1001-1 Kishioka-cho, Suzuka 510-0293, Mie, Japan

**Keywords:** Eucommia leaf extract, antihypertensive, anti-obesity, asperuloside, geniposidic acid, ANP secretion, gut microbiota, short-chain fatty acids, rodents, humans

## Abstract

In the 1970s, Eucommia leaf tea, known as Tochu-cha in Japanese, was developed from roasted Eucommia leaves in Japan and is considered as a healthy tea. The antihypertensive, diuretic, anti-stress, insulin resistance improving, and anti-obesity effects of Eucommia leaf extract have been reported. However, the identification and properties of the active components as well as the underlying mechanism of action are largely unknown. In this review, we summarize studies involving the oral administration of geniposidic acid, a major iridoid component of Eucommia leaf extract which increases plasma atrial natriuretic peptide (ANP) on the atria of spontaneously hypertensive rats (SHR) by activating the glucagon-like peptide-1 receptor (GLP-1R). To achieve the antihypertensive effects of the Eucommia leaf extract through ANP secretion in humans, combining a potent cyclic adenosine monophosphate phosphodiesterase (cAMP-PDE) inhibitor, such as pinoresinol di-β-d-glucoside, with geniposidic acid may be necessary. Changes in the gut microbiota are an important aspect involved in the efficacy of asperuloside, another component of the Eucommia leaf extract, which improves obesity and related sequelae, such as insulin resistance and glucose intolerance. There are species differences of mechanisms associated with the antihypertensive and anti-obesity effects between rodents and humans, and not all animal test results are consistent with that of human studies. This review is focused on the mechanisms in antihypertensive and anti-obesity effects of the Eucommia leaf extract and summarizes the differences of mechanisms in their effects on rodents and humans based on our studies and those of others.

## 1. Introduction

*Eucommia ulmoides* Oliv. is a deciduous tree that can grow up to 20 m in height. The bark of the tree has been listed in the Chinese pharmaceutical book “Shennong Ben Cao Jing” since 300 AD. The Chinese pharmaceutical book Compendium of Materia Medica, which was published in the late 16^th^ century, also lists its effects and usage, including its antihypertensive, diuretic, tonicity, and analgesic activities as a traditional medicine, Eucommia Bark. On the other hand, Western Chinese folklore reports that Eucommia leaves were only used for preparing tea and porridge by minorities, such as the Qiang people.

In the 1970s, Eucommia leaf tea, known as Tochu-cha in Japanese, was developed from roasted Eucommia leaves in Japan and is considered a healthy tea. The antihypertensive and anti-obesity effects of the Eucommia leaf extract (ELE) have resulted from rodent animal test results; however, the details of the active components in ELE and the mechanisms of action are not known. The main components in ELE are geniposidic acid (GEA), chlorogenic acid (CA), and asperuloside (ASP) (Figure 1) [1].

We have conducted studies on the antihypertensive and anti-obesity effects of ELE and its active components. So far, nobody has reported on the antihypertensive and anti-obesity effects of ELE in human clinical trials; we only have the reports of our research collaborators. In this process, we reported for the first time that there are species differences of mechanisms associated with the antihypertensive and anti-obesity effects between rodents and humans, and not all animal test results are consistent with that of human studies.

This review focuses only on the antihypertensive and anti-obesity effects of ELE and summarizes the differences of mechanisms in their effects between rodents and humans based on our studies and those of others.

## 2. The Contents of The Main Components in Eucommia Leaf Extract

We analyzed an aliquot of ELE produced from Eucommia leaves cultivated in Japan and China based on a content of 85 mg GEA and measured the amount of CA and ASP in the extracts. CA and ASP were present at 48.6–75.7 mg and 14.8–21.6 mg, respectively, in 1.20–1.33 g of ELE extract [1]. The activity of CA from coffee beans has been analyzed in rats and humans with respect to its antihypertensive and anti-obesity effects. The minimum active amount of CA in a human clinical trial was reported to be 185 mg daily to assess antihypertensive effects [2], and 297 mg daily for determining its anti-obesity effects [3]. In addition, the minimum amount of CA in animal studies was reported to be 300 mg/kg/day in rats for antihypertensive effects [4], and 100 mg/kg/day in mice for anti-obesity tests [5]. The content of CA in ELE (48.6–75.7 mg/85 mg GEA) is too small to demonstrate antihypertensive or anti-obesity effects. Therefore, we concluded that the effects observed with ELE in both humans and rats are likely not due to CA. Therefore, this review is focused on the antihypertensive and anti-obesity effects resulting from GEA and ASP in ELE.

## 3. Atrial Natriuretic Peptide Secretion by Geniposidic Acid in Eucommia Leaf Extracts in Rats

GEA has pharmacological effects on hypertension, inflammation, diabetes, and arteriosclerosis. Metabolic analyses have been performed following the oral administration of active components of Eucommia leaves, including iridoids; however, the active components and underlying mechanisms have not been elucidated [6]. We evaluated the mechanism underlying the antihypertensive effects of a single oral dose of 100 mg GEA/kg from ELE in spontaneously hypertensive rats (SHR) [7]. Based on the marked antihypertensive effects of GEA, we hypothesized that GEA may promote atrial natriuretic peptide (ANP) secretion by acting as a glucagon-like peptide-1 receptor (GLP-1R) agonist [8].

Changes in systolic blood pressure (SBP) were calculated from the values measured before and after oral administration. Initial SBP values for the control group and before oral administration of 50 and 100 mg/kg GEA were 196.3 ± 5.6, 204.1 ± 6.4 and 203.2 ± 2.0 mmHg, respectively. At 6 h post-administration, SBP decreased significantly in rats treated with 50 and 100 mg/kg GEA (166.6 ± 2.6 and 153.0 ± 2.7 mmHg, respectively) compared with the controls (183.6 ±7.2 mmHg). Based on these findings, we examined the effects of 100 mg/kg on SBP and heart rate (HR) and measured the circulating levels of GEA and ANP 6 h following administration. SBP and HR were significantly reduced in the GEA-treated group compared with the control group (*p* < 0.01 for each parameter). Plasma ANP levels were significantly increased in the GEA-treated group (69.4 ± 5.2 pM), which is up to 50 % of the level observed in the control group (46.2 ± 5.3 pM) (*p* < 0.01).

The mechanism by which GLP-1R agonists exert antihypertensive effects was unknown. Recently, Kim et al. demonstrated that ANP was essential for the GLP-1-stimulated relaxation of vascular smooth muscles [9]. After the activation of atrial GLP-1Rs by an agonist, such as liraglutide, the increased cyclic adenosine monophosphate (cAMP) levels promote membrane translocation of exchange proteins directly activated by cAMP, which subsequently mediates ANP release from the large dense core vesicle. ANP induces guanosine 3′,5′-cyclic monophosphate (cGMP)-mediated smooth muscle relaxation, resulting in a reduction in blood pressure. The GLP-1R blockade by exendin (9–39) reduced GLP-1R agonist-induced ANP secretion. Therefore, it is clear that as a GLP-1R agonist, GEA increases ANP secretion via GLP-1R activation and the exchange protein activated by cAMP (Epac 2) translocation [7].

Kwan et al. conducted animal experiments and informed us privately that GEA did not produce direct vasodilation of the isolated rat aorta. In addition, GEA produced very little endothelium-dependent relaxation of the isolated rat aorta, even at very high concentrations (private note, unpublished data). These findings indicate that GEA acts as a GLP-1R agonist and increases ANP secretion, thus reducing SBP in SHRs.

Our study provided the first evidence that oral administration of GEA increased plasma ANP by activating GLP-1R in a rat model of hypertension as an iridoid component isolated from a natural medicinal plant, unlike peptide agonists [7]. This GEA as the agonist appeared to have favorable effects on cardiovascular risk factors, such as blood pressure and lipid levels; however, efficacy has not yet been demonstrated for natural products [10]. Based on results with liraglutide, GEA may act as a GLP-1R agonist and promote the secretion of ANP in SHR [9]; however, ANP secretion by GEA alone and antihypertensive effects have not been observed in humans [11].

Skov et al. reported that ANP secretion by GLP-1R agonists and antihypertensive effects in humans, unlike rodents, were not observed, suggesting that the GLP-1-ANP axis is absent in humans [12]. In fact, GEA promotes natriuresis, but promotes neither ANP secretion nor antihypertensive effects in humans [11], similar to that observed with liraglutide [13]. Nakagami et al. indicated that increased cAMP levels are required to induce membrane translocation of the exchange proteins directly activated by Epac 2 [14]. In addition, there are species differences in antihypertensive effects between rodents and humans [11]. GEA secretes ANP in rodents, but not in humans, and may only exhibit an increase of urinary sodium excretion independent of antihypertensive effects, similar to that observed with liraglutide [13].

In contrast, Greenway et al. reported that an aqueous Eucommia bark extract exhibited an antihypertensive effect in a human clinical trial [15]. The extract was standardized to 8% pinoresinol di-β-d-glucoside (PDG) for blood pressure reduction in humans and an apparent reduction of HR over 24 h by ambulatory blood pressure monitoring (ABPM) was observed [15]. The results of this Eucommia bark extract trial is consistent with the effects observed using a β-adrenergic blocker [11]. Examining the effect of Eucommia bark extract on isoproterenol-stimulated lipolysis using a human fat cell assay confirmed that the extract does indeed exhibit β-adrenergic blocking activity. Thus, Eucommia bark extract may promote the secretion of ANP and act as a β-adrenergic blocker in humans, unlike ELE [15].

One of the main components of the Eucommia bark extract is GEA, which is the same as that of ELE. In addition, Eucommia bark extract contains PDG as another major component (Figure 2), which is barely present in ELE. It is known that PDG exhibits high cyclic adenosine monophosphate phosphodiesterase (cAMP-PDE) inhibitory activity, IC_50_ (×10^−5^ M).

## 4. The Antihypertensive Effect of Eucommia Leaf Extract in Rodents

### 4.1. Eucommia Leaf Extract—Hypertension

We examined the antihypertensive effects of ELE (2000 mg/kg) following a single *p.o.* administration to SHR. The ELE-treated group experienced significant antihypertensive effects at 3, 6, and 9 h after administration compared with the controls; however, the antihypertensive effect tended to diminish 24 h after administration (Figure 3) [16].

### 4.2. Geniposidic Acid-Atrial Natriuretic Peptide

For a long time (almost 20 years or more), it was believed that the antihypertensive effects of ELE depended on the direct action of GEA on the parasympathetic nervous system, which results in the release of nitric oxide (NO) from vascular endothelial cells, followed by the relaxation of vascular smooth muscle [17]. However, this hypothesis is not correct because GEA does not act on the muscarinic acetylcholine receptor [18]. We clarified that the antihypertensive effect of ELE in rodents depends on ANP secretion by GEA, as shown in Figure 4 [11].

Imaizumi et al. demonstrated that ANP inhibits sympathetic ganglionic transmission and enhances the effect of cardiac parasympathetic nerve activity on HR [19]. In addition to direct vasodilating and renal effects, ANP has an important role in cardiovascular regulation by affecting sympathetic nerve activity and HR. We observed that the oral administration of GEA to SHR increased blood levels of ANP, lowered blood pressure, and decreased HR [7]. Pennacchio et al. reported that GEA decreased myocardial contractility and HR in Langendorff experiments using isolated Wistar rat hearts [20]. We observed that the intravenous administration of ELE containing GEA to SHR immediately decreased HR and lowered blood pressure, whereas the administration of atropine suppressed HR and blood pressure by 66.5% [16]. This indicates that the enhancement of parasympathetic nerve activity with respect to HR is suppressed by atropine.

### 4.3. Atrial Natriuretic Peptide-Nitric Oxide Pathway

Costa et al. studied the relationship between the antihypertensive effects of ANP and the NO pathway in male Wistar rats injected with saline [21]. They demonstrated that NG-nitro-l-arginine methyl ester (l-NAME) reverted the decrease in mean arterial pressure induced by ANP administration. This suggests that ANP increases NO synthesis capability in vascular smooth muscle cells and endothelial cells, in which the cGMP pathway may be involved [21]. The cGMP signaling pathway is believed to be an important regulator of cardiovascular and renal physiology. Endothelial nitric oxide synthase (eNOS) activity stimulated by ANP in the kidney, aorta, and left ventricle, was partially abolished by the natriuretic peptide receptor (NPR)-A/B) antagonist as well as protein kinase G (PKG) inhibition. ANP interacts with NPR-A/B and increases cGMP, which, in turn, activates PKG to stimulate eNOS [22]. Therefore, the NO pathway could be an intercellular messenger in the ANP endothelium-dependent vasorelaxation mechanism and the activation of the NO pathway may be one mechanism involved in the diuretic, natriuretic, and antihypertensive effects of ANP [22]. Recently, Ishimitsu et al. reported that the increased expression of eNOS and the increased bioavailability of NO ameliorates hypertension and improves renal hemodynamics with ELE or GEA in male Dahl salt sensitive rats administered drinking water containing 1% salt [23].

We measured SBP and the effect of ELE on plasma NO in the thoracic aorta of Wistar Kyoto rats (WKY) and SHR. SBP was measured by the tailcuff method and the data were expressed as the change in SBP during three- and seven-week studies (ΔSBP) (Table 1) [24]. Plasma NO levels were used to evaluate the improvement in endothelial function following ELE treatment (Table 2). NO levels in the SHR-ELE group were significantly increased compared with that in the SHR-control group. We evaluated the long-term effects of ELE on endothelial function by measuring the aortic media thickness. The arterial media thickness was significantly decreased in the SHR-ELE group compared with that in the SHR-control group (Table 2). Long-term ELE administration may effectively improve vascular function by increasing plasma NO levels and bioavailability and by decreasing arterial media thickening in SHR, followed by an antihypertensive effect in SHR. Based on the vasorelaxation mechanisms resulting from the direct action on the smooth muscle of the aorta and the NO pathway in the endothelium, the observed antihypertensive effect of ELE treatment in rodents may depend on ANP secretion induced by GEA.

## 5. The Anti-Obesity Effect of Eucommia Leaf Extract in Rodents

### 5.1. Eucommia Leaf Extract—Obesity

We established a metabolic syndrome-like model by feeding rats with a 35% high-fat diet (HFD) to examine potential anti-obesity and anti-metabolic syndrome effects following the chronic administration of ELE [25]. Food intake was significantly decreased in the ELE treatment group compared with the control group. It is known that the food intake directly affects the obesity. It seems that the food intake might decrease with the decrease in the body weight. However, our studies of the ELE treatment group on physical parameters white adipose tissue (WAT weight) and plasma parameters (adiponectin) as well as gene expression analysis by real-time PCR in adipose tissue clearly indicated the high anti-obesity effect by the energy expenditure in the adipose tissue (Table 3 and Table 4). The anti-obesity effect of the ELE treatment group is not simply due to the weight loss by reduced food intake. In both studies, especially increases of adiponectin, an adipocyte-derived hormone was observed following ELE administration. Adiponectin is potentially an important therapeutic to reduce the burden associated with obesity and related chronic diseases [26].

### 5.2. Asperuloside

We established a metabolic syndrome model by feeding rats with a 35% HFD and examined potential anti-obesity and anti-metabolic syndrome effects by the chronic administration of ASP [27]. A total of six rats were divided into three groups and studied for three months. ASP suppressed body weight, visceral fat weight, food intake, and circulating levels of glucose, insulin, and free fatty acids, whereas plasma adiponectin levels were increased in rats on an HFD-ASP (Table 5). RT-PCR studies also showed the anti-obesity and anti-metabolic syndrome effects from the chronic administration of ASP (Table 6).

Nakamura et al. determined the mechanism by which ASP improves metabolic health in mice following the oral administration of 0.25% ASP in HFD [28]. The results indicated that changes in gut microbiota are an important aspect involved in the effects of ASP with regard to improving the obesity and associated phenotypes, such as insulin resistance and glucose intolerance. ASP alters gut microbiota at the phylum level with an important impact on the ratio of *Bacteroidetes* to *Firmicutes.* This is consistent with previous studies indicating that microbiota in obese mice are enriched in *Firmicutes* and decreased in *Bacteroides* [28]. A similar effect was observed in HFD-fed mice, in which ASP-administration decreased the amount of *Firmicutes* and increased that of *Bacteroidetes*. In addition, at the genus level, ASP increased the amount of *Parabacteroides*. These findings suggest that the gut microbiota are an important target for both the prevention and treatment of obesity and metabolic dysfunction. *Parabacteroides*, *Roseburia*, and *Anaerostipes* produce short-chain fatty acids (SCFAs).

SCFAs were significantly increased by ASP treatment. SCFAs are important secondary metabolites that activate certain G protein-coupled receptors (GPR 41/43) and enhance metabolite signaling. GPR41 activation by SCFAs (butylic acid) increases noradrenaline release from sympathetic neurons, whereas GPR43 activation by SCFAs (acetic acid) suppresses insulin-mediated fat accumulation in white adipose tissue (WHT). [29]. Nakamura et al. reported that *Akkermansia muciniphila* was markedly increased in ASP-treated mice. *Akkermansia muciniphila* is considered a beneficial microbe which produces active SCFAs from the dietary fiber in the gut, which results in anti-obesity effects [28].

ASP may be one of the bioactive compounds responsible for the anti-obesity effects of ELE, and we have suggested an additional mechanism by which ASP treatment ameliorates obesity in addition to the one reported by Nakamura et al. [30]. It has long been known that the leaves of *Eucommia* are hardly damaged by insects, and ASP is an insect repellent [31]. The related iridoids glucoside, gardenoside from the gardenia fruit is also known as an insect repellent. Gardenoside is easily hydrolyzed to aglycone in the digestive tract by β-glucosidase, followed by its binding to transporter proteins to cause lethal damage [31,32].

The bioavailability of ASP is very low and plasma levels are extremely low in rats (Cmax = 198 ng/mL at 50 mg/kg *p.o*.). ASP does not readily enter the blood. Once it reaches the ileum, it is hydrolyzed to an unstable aglycone by the intestinal bacteria-derived β-glucosidase [30,31]. It is known that the unstable aglycone from ASP readily binds with proteins, similar to gardenoside [30,31]. Therefore, an increase in the bile acid pool in the gut following ASP administration in rats may be attributable to a reduction of ileal absorption of bile acids by the transporter bound with the aglycone of ASP. Thus, it is likely that the anti-obesity effect of ASP in rodents may be an indirect effect of cholic acid, rather than the direct action of ASP.

A similar mechanism of increased bile acids in the ileum has been reported for metformin, which is derived from the natural product phenformin, by chemical modification. Metformin is used as a first line therapy for type 2 diabetes. Large cohort studies have shown weight loss benefits associated with metformin therapy [33]. Metformin inhibits the farnesoid X receptor (FXR) via an AMP-activated protein kinase (AMPK)-mediated mechanism, which results in reduced sensing and the ileal absorption of bile acids, followed by an increase in the bile acid pool of the ileum [34]. Although the bile acid inhibition pathways of ASP and metformin are different, they both enhance the bile acid pool in the ileum.

The reported results of ASP treatment are consistent with that reported for metformin with respect to changes in the relative concentration of *Bacteroides* and *Firmicutes* [35], the modulation of gut microbiota, the increase of SCFAs metabolizing bacteria, and the increased abundance of *Ackermansia* [36]. Based on these findings, we present a mechanism for the amelioration of obesity in mice by ASP treatment (Figure 5) [30].

### 5.3. Geniposidic Acid

Bordicchia et al. reported that activation of the β-adrenergic receptor (β-AR) can induce a functional “brown-like” adipocyte phenotype [37]. As ANP and β-AR agonists are similarly potent at stimulating lipolysis in human adipocytes, they determined whether ANP could induce human and mouse adipocytes to acquire brown adipocyte features, including a capacity for thermogenic energy expenditure mediated by uncoupling protein 1 (UCP1). In human adipocytes, ANP activates PPARγ coactivator-1α (PGC-1α) and UCP1 expression, induced mitochondrial biogenesis, and increased uncoupled and total respiration. In both human and mouse adipocytes, ANP-treatment resulted in the same increase in PGC-1α and UCP1 expression, mitochondrial content, and uncoupled respiration as observed with β-agonists, all in a p38 mitogen-activated protein kinase (MAPK)-dependent manner. These results suggest that ANP promotes the “browning” of white adipocytes to increase energy expenditure, which defines the heart as a central regulator of adipose tissue biology. It was reported that ANP and β-agonists can work together through p38 MARK signaling to generate additive or synergistic effects. For clinical applications, Bordicchia et al. suggested that the ability of ANP together with catecholamines to modulate uncoupled respiration and control white fat mass may serve as a strategy to manage obesity and the metabolic complications associated with it.

Recently, Kimura et al. also evaluated the effect of ANP-treatment on adipose tissue browning and thermogenesis in mice [38]. Mice fed a high-fat diet (HFD) or a normal-fat diet (NFD) for 13 weeks were untreated or administered ANP subcutaneously for an additional 3 weeks. ANP-treatment significantly ameliorated HFD-induced insulin resistance. HFD increased brown adipose tissue (BAT) cell size along with the accumulation of lipid droplets, which was suppressed by ANP-treatment. Furthermore, HFD induced enlarged lipid droplets in WAT and hepatic steatosis, all of which were markedly attenuated by ANP-treatment. We also observed similar results in male Sprague–Dawley (SD) rats following the oral administration of ELE [39].

Kimura et al. indicated that ANP-treatment markedly increased UCP1 expression, a specific marker of BAT, in WAT browning and UCP1 expression in BAT with NFD [38]. Accordingly, a cold tolerance test demonstrated that ANP-treated mice were tolerant to cold exposure. Kimura et al. suggests that the administration of agents which increase circulatory ANP levels may have therapeutic benefits based on these results. GEA, which promotes ANP secretion following oral administration, may represent a novel therapeutic agent for metabolic syndromes. This additional mechanism of action for ELE in rats to improve obesity by GEA is presented in Figure 6.

From the review of our work and that of others, we conclude that the amelioration of obesity by ELE in rodents may depend on additive or synergistic effects of combinations, such as β-agonist activation by SCFAs with ASP treatment (Figure 5) and ANP secretion with GEA treatment (Figure 6). Therefore, ELE may serve as an effective treatment for the management of metabolic disorders, such as hypertension, as well as obesity and diabetes.

### 5.4. The Blood Sugar Lowering Effect of Asperuloside Treatment

Nakamura et al. observed that incretin concentrations were higher in ASP-treated mice compared with that in HFD groups. Incretin stimulates insulin secretion from pancreatic β-cells in a glucose-dependent manner, whereas increasing and stabilizing incretin levels represents a useful strategy to manage type 2 diabetes [28]. Recently, SCFAs were shown to activate the gut receptors, GPR41 and GPR43, to stimulate L-cells to release incretin. Nakamura et al. produced in vitro data showing the absence of a direct effect of ASP on incretin release in cultured NCI-H716L cells, which is consistent with an indirect effect mediated by increased SCFAs levels. We presumed that one of the blood sugar lowering effects of ASP in rodents may depend on an indirect effect of cholic acid, not on the direct effects of ASP; this being the increase of bile acid pool in the ileum due to the inhibition of bile acid transporter by asperuloside [28,30,31] (Figure 7).

## 6. The Antihypertensive Effect of Eucommia Leaf Extract in Humans

The antihypertensive effects of beverages containing ELE (80 mg as GEA) were examined in a randomized, double-blind, placebo-controlled, parallel group study, which was reported by Tsukamoto et al. [40]. The subjects included high normotensive adult male and female subjects. Test or placebo meals were provided once a day for 12 weeks. Significant differences in SBP were observed at 6, 8, 10, and 12 weeks. While diastolic blood pressure (DBP) values were significantly different at 8, 10, and 12 weeks, no changes in BP were observed in the placebo group (Figure 8).

ELE exhibited an antihypertensive effect following oral administration to humans [40]. However, since ANP secretion was not observed in humans, we discussed the possibility of mechanisms other than ANP secretion [1]. Kario reported that many hypertensive factors are associated with the increased inflammatory response, and inflammation causes damage to the vascular endothelium [41]. The administration of tumor necrosis factor α (TNFα) inhibitors protects blood vessels and directly suppresses inflammation [41].

Faria et al. observed the antihypertensive effects of infliximab, a TNF-α inhibitor used for rheumatoid arthritis, in a randomized, double-blind, placebo-controlled pilot study in resistant hypertensive subjects [42]. Hürlimann et al. showed that an inflammatory response pathway in common with that of rheumatoid arthritis is involved in the pathogenesis of arteriosclerotic vascular diseases, and that the administration of infliximab for 12 weeks markedly improved flow-mediated vasodilation (FMD). (3.2 ± 0.4% to 4.1 ± 0.5%, *p* = 0.018) [43].

Filho. et al. reported that the antihypertensive effect of infliximab is mediated by the Akt/eNOS (Ser 1177) pathway [44]. Both infliximab and GEA are TNFα inhibitors and have anti-rheumatic and anti-inflammatory effects.

Xing et al. reported that ELE inhibited interleukin IL-6, IL-17, and TNF-α mRNA expression in the spleen of collagen-induced arthritis rats [45]. We showed that the administration of ELE to rats on a high-fat diet reduced TNFα levels in the blood [25]. Furthermore, we reported the suppression of vascular inflammation in ApoE KO mice through TNFα inhibition by ELE and the protection against atherosclerosis [46]. These findings suggested that ELE has an anti-inflammatory effect resulting from TNFα inhibition. Kim et al. also reported that Eucommia leaves represent a traditional medicine used for rheumatoid arthritis in South Korea, and that they have a higher inhibitory effect on prostaglandin E2 (PGE2), IL-6, and TNFα compared with indomethacin [47]. Jin et al. reported that GEA, the major component, reduces serum TNFα levels in adjuvant-induced arthritis rats [48]. Gao et al. reported the anti-arteriosclerotic effect of GEA in rabbits and the improvement of vascular endothelial cell function using human umbilical vein endothelial cells (HUVECs) [49]. Based on these reports, ELE and GEA were also presumed to have anti-inflammatory effects on vascular endothelial cells.

Chang et al. also reported that A *Torenia comcolar* Lindley var. *formosama* Yamazaki ethanolic extract has anti-inflammatory properties due to TNFα inhibition via the Akt/eNOS (ser 1177) pathway, similar to infliximab [50].

In the case of the antihypertensive effect due to the anti-inflammation by GEA in the animal trials in vivo using rodents, the antihypertensive effect due to the anti-inflammation might be difficult to distinguish from the antihypertensive effect masked by the action due to the ANP. This is because of the antihypertensive effect due to the ANP secretion that is observed soon after the single oral administration of ELE [16]. Therefore, as a preliminary trial, we compared the similarity of the onset time of the significant antihypertensive effect without the ANP secretion after the initiation of oral administration of ELE between the reported data of infliximab and our data of GEA in human clinical trials [1].

Our research collaborator, Hosoo et al., performed a randomized, double-blind, placebo-controlled, paralleled group study. The subjects were 46 adults (28 males and 18 females) with high normal blood pressure. They administered 350 mL of a beverage containing 85.0 mg of GEA to human subjects once a day for 12 weeks and measured the blood pressure and the change in %FMD. The onset time of the significant antihypertensive effect (four weeks after the initiation of oral administration) and an improvement in FMD (eight weeks after the initiation of oral administration) of 1.29 ± 1.65% (compared with pre-intake values) were observed in the treated group compared with the placebo control group of −0.01 ± 1.84%, respectively [51].

These results of the GEA suggested the similarity with those of infliximab as a TNFα inhibitor and the improvements of both blood pressure and FMD by Akt activation—eNOS-derived NO increase and vasodilatation [1]. Furthermore, we need a more detailed evaluation of the mechanism of the antihypertensive effect of GEA in humans in the future.

## 7. The Anti-Obesity Effects of Eucommia Leaf Extract in Humans

In the case of the anti-obesity effect of ELE in humans, ANP secretion is not observed, unlike in rodents. Therefore, the improvement of obesity in humans depends primarily on the effects of ASP. We hypothesized that the effects of ASP in humans depends on an indirect effect of cholic acid, the modulation of gut microbiota, and an increase of SCFAs metabolizing bacteria, similar to that of metformin [30].

Because of the modest and inconsistent effects of weight loss, the FDA has not approved metformin as a weight loss agent [33]. Dietary fiber in the gut is needed to produce active SCFAs for the anti-obesity effects of ASP [34] and metformin [52]. We conducted a clinical trial on human subjects to determine the anti-obesity effect of ELE; however, we did not observe significant anti-obesity effects other than body fat reduction in Japanese subjects [53]. In contrast, the anti-obesity effects were observed in Chinese subjects by Zhou et al., and our research collaborators, Hirata et al. [54]. The human clinical trials were carried out in China using the ELE prepared by Kobayashi Pharmaceutical Company in Japan, which is equivalent to the clinical trials in the Japanese subjects.

Twenty-seven subjects with abdominal obesity were randomly divided into two groups (an experimental group of 15 subjects and a control group of 12 subjects), and they were orally administrated an a Eucommia capsule containing 1500 mg of ELE or a placebo capsule. Changes in weight, abdominal circumference, body mass index (BMI), visceral fat, and subcutaneous fat in each group were measured after 8 weeks of ELE treatment. Compared with the control group, visceral fat (270.1→214.7 cm^2^) and subcutaneous fat (263.1→235.6 cm^2^) in the experimental group decreased significantly (*p* < 0.05), weight (97.2→95.9 kg) and abdominal circumference (109→106 cm) decreased (*p* < 0.001), and BMI (31.7→31.3 kg/m^2^) also decreased (*p* < 0.01).

It is known that ELE exhibited an anti-obesity effect when the extract was taken with a meal containing dietary fiber, such as high β-glucan barley [55]. This indicates that ASP also requires dietary fiber to produce significant anti-obesity effects similar to that observed with metformin. The combination of dietary fiber and metformin was shown to have a synergistic effect and was more effective than dietary fiber or metformin alone at improving obesity and insulin resistance in a human clinical trial [52].

It is known that the intake of vegetables rich in dietary fiber among Chinese subjects is the highest in the world, while that of Japanese subjects is considerably lower.

We hypothesized that the difference in the anti-obesity effect between Japanese and Chinese subjects may be attributable to the difference of dietary fiber intake. The results of rodent animal studies evaluating anti-obesity effects does not necessarily apply to that in humans, as there are no additive or synergistic effects observed with ANP secretion.

## 8. Conclusions

We found that there are species differences with regard to the mechanisms of antihypertensive and anti-obesity effects between rodents and humans, and not all rodent animal test results are fully applicable to human clinical trial results. We reported for the first time that the oral administration of GEA, which is a medicinal plant component rather than a peptide agonist, increased plasma ANP secretion by activating GLP-1R on the atria of SHR. However, we did not observe ANP secretion by oral administration of ELE containing GEA alone in humans. This suggests that the ANP secretion may occur using a combination of potent cAMP-PDE inhibitors, such as PDG, with GEA in humans.

In this review, we summarized that ELE, GEA and ASP exhibit a potential benefit for the management of metabolic disorders, hypertension, obesity, and diabetes in rodents. However, to demonstrate these effects in the case of humans, a combination of a potent cAMP-PDE inhibitor with ELE may be necessary, which requires a further human clinical evaluation in the future.

## Figures and Tables

**Figure 1 molecules-28-01964-f001:**
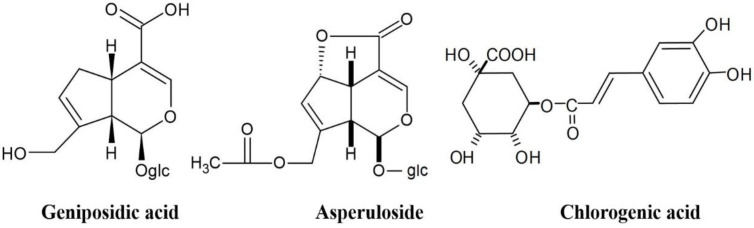
Chemical structures of the main components of ELE.

**Figure 2 molecules-28-01964-f002:**
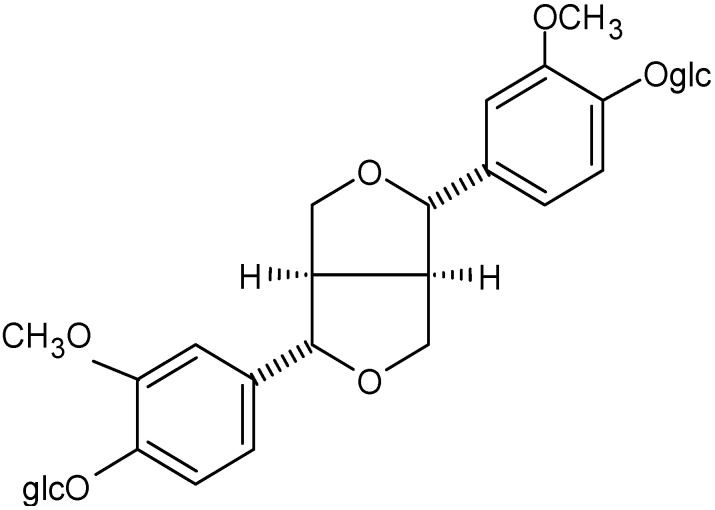
Chemical structure of PDG.

**Figure 3 molecules-28-01964-f003:**
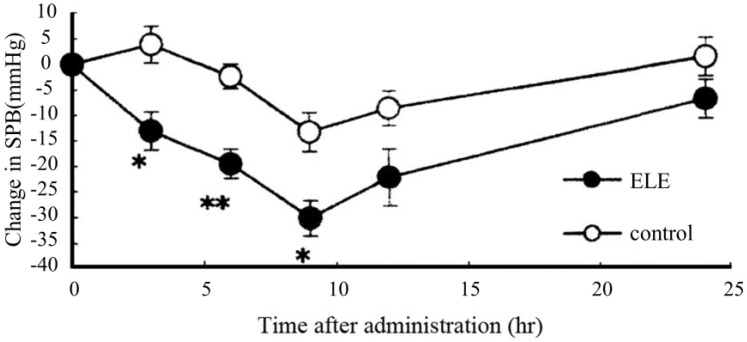
Effects of a single oral administration of ELE in SHR. Change in SPB is expressed as the difference in systolic blood pressure (SPB) before and after administration. ●: ELE (2000 mg/kg); ○: saline as the control. Each value represents the mean ± standard error (S.E.) (*n* = 5–6), *n* per group: ELE = 5 and control = 6. * *p* < 0.05 and ** *p* < 0.01 (Student *t*-test).

**Figure 4 molecules-28-01964-f004:**
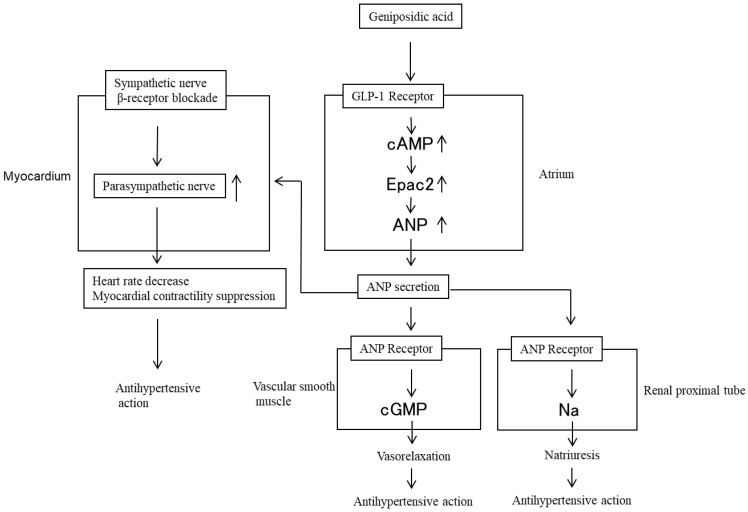
Antihypertensive mechanism of action of ELE resulting from atrial natriuretic peptide (ANP) secretion by GEA in rodents; cGMP: guanosine 3′,5′-cyclic monophosphate; cAMP: cyclic adenosine monophosphate. Small arrows next to words indicate the increases or decreases in the action or secretion.

**Figure 5 molecules-28-01964-f005:**
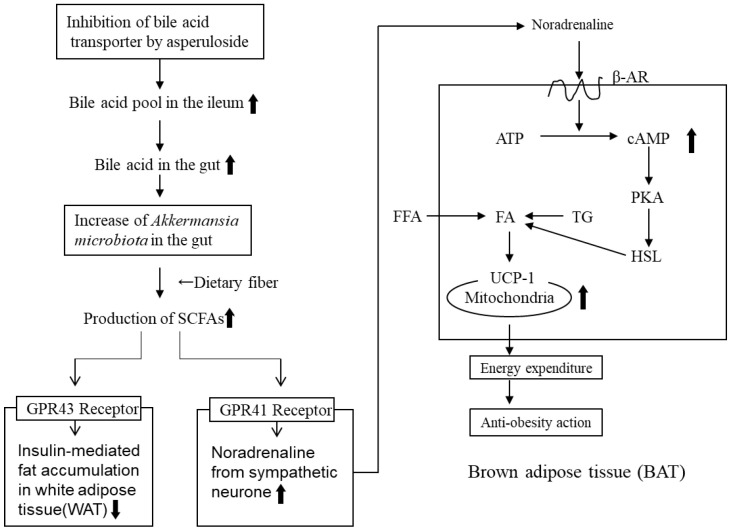
Mechanism of the anti-obesity effects of ELE-containing ASP via SCFAs. SCFAs: short-chain fatty acids, GPR: G protein-coupled receptors, β-AR: β-adrenergic receptor, PKA: protein kinase A, HSL: hormone sensitive lipase, TG: triglyceride, FA: fatty acid, FFA: free fatty acid. WAT: white adipose tissue. Small arrows next to words indicate the increases or decreases in the action or secretion.

**Figure 6 molecules-28-01964-f006:**
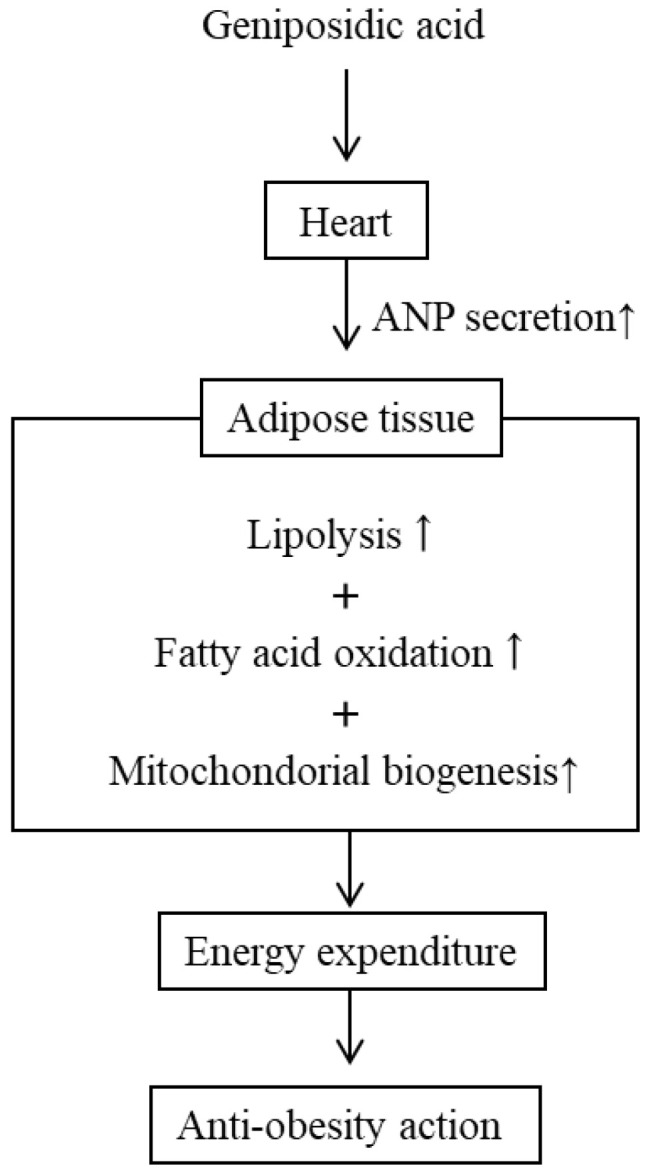
Anti-obesity mechanism of action of GEA through ELE treatment by ANP secretion in rodents. Small arrows next to words indicate the increases or decreases in the action or secretion.

**Figure 7 molecules-28-01964-f007:**
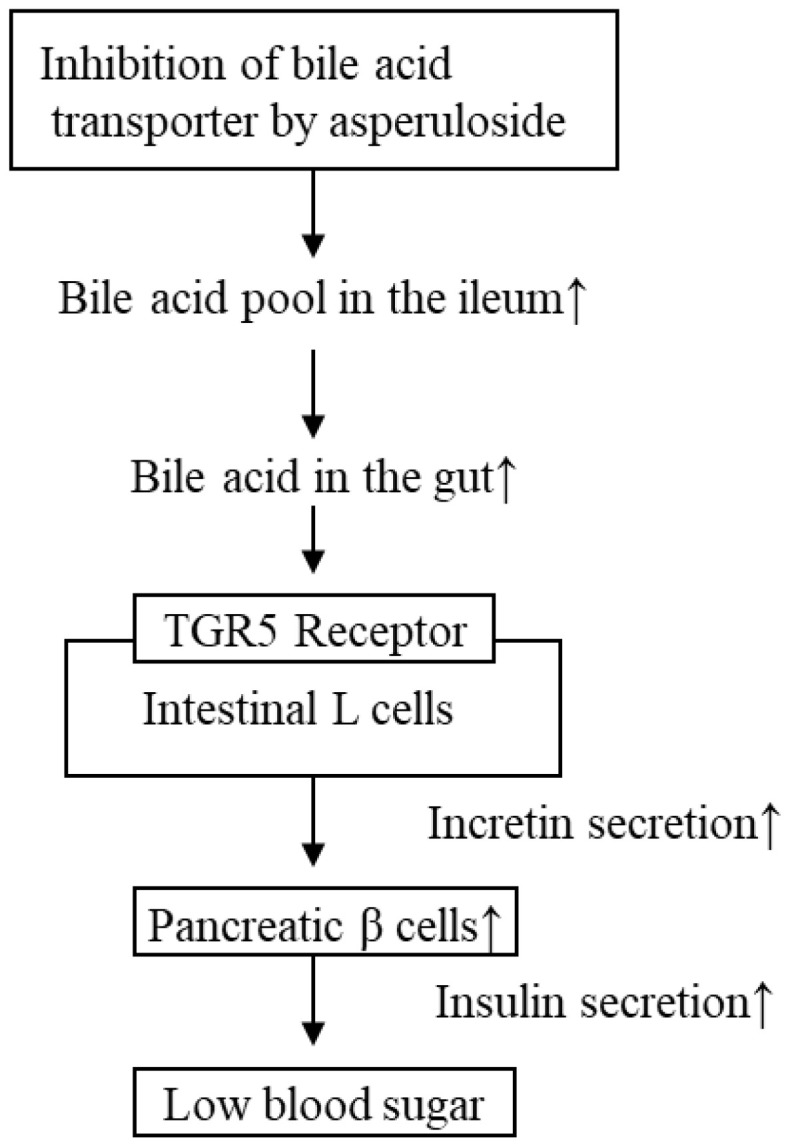
Mechanism of action for the blood sugar lowering effects of ASP. TGR5 Receptor: transmembrane G protein-coupled receptor 5. Small arrows next to words indicate the increases or decreases in the action or secretion.

**Figure 8 molecules-28-01964-f008:**
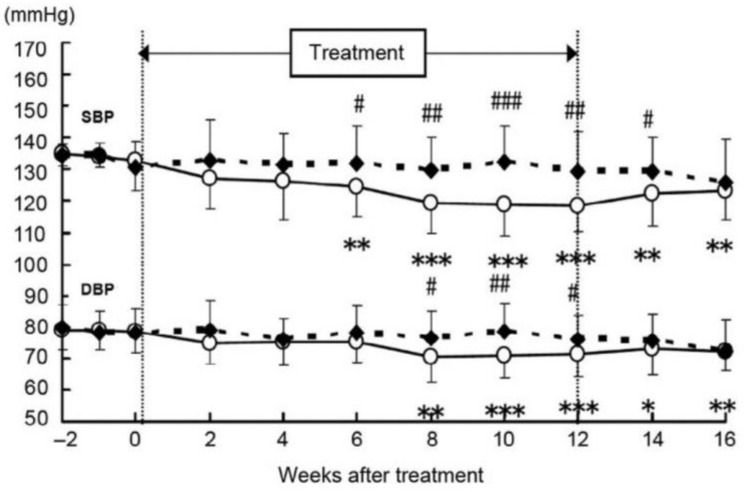
Change in the systolic blood pressure (SBP) and diastolic blood pressure (DBP) in high normotensives humans. ○: test beverage group *(n* = 21), ◆: placebo group (*n* = 22). The values are expressed as the mean ± S.E. Significant differences are as follows: ^#^ *p* < 0.05, ^##^ *p* < 0.01, ^###^ *p* < 0.001, compared with placebo-meal (control) group using a Student’s *t*-test; * *p* < 0.05, ** *p* < 0.01, *** *p* < 0.001, compared with values on day 0 after intake (initiation day) using Bonferroni’s multiple rank-sum test.

**Table 1 molecules-28-01964-t001:** The change in systolic blood pressure (SBP) during the seven-week study.

	ΔSBP (mmHg)	
	Week 3	Week 7
WKY SHR-control SHR-ELE	3.6 ± 1.8 ** 35.6 ± 0.7 17.8 ± 2.5 **	16.4 ± 4.7 ** 72.9 ± 2.4 67.1 ± 2.3 **

The ∆SBP is the SBP at the indicated time point minus the SBP on day 1. All values represent the mean ± S.E. (*n* = 7), ** *p* < 0.01, compared with the SHR-control group (Dunnett’s test).

**Table 2 molecules-28-01964-t002:** Plasma NO levels and thoracis aorta ring media thickness in the indicated treatment groups.

	Plasma Nitric Oxide Level (µM)	Media Thickness (µm)
WKY	3.49 ± 0.22	90.4 ± 3.6 **
SHR-control	6.37 ± 0.59	124.7 ± 5.9
SHR-ELE	11.75 ± 1.95 *	101.9 ± 1.4 **

The values represent the means ± S.E. (*n* = 7); * *p* < 0.05, ** *p <* 0.01 as compared with SHR-control groups (Dunnett’s test).

**Table 3 molecules-28-01964-t003:** Effects of ELE on physical and plasma parameters after a three-month administration in HFD-fed rats.

		Diet (HFD)	
	Control	3% ELE	9% ELE
Physical parameters			
Final body weight (g/rat)	548.6 ± 16.8	485.3 ± 13.6 *	422.2 ± 17.7 *
Food intake (g/day/rat)	25.3 ± 3.0	19.7 ± 3.6	15.2 ± 1.7 *
WAT weight (g/rat)			
Perirenal white adipose tissue	10.0 ± 0.8	5.6 ± 0.4 ***	3.5 ± 0.6 ***
Epididymal white adipose tissue	18.3 ± 0.7	13.9 ± 0.6 ***	5.8 ± 0.4 ***
Plasma parameters			
Glucose (mg/L)	1520 ± 17	1458 ± 5 *	1433 ± 17 *
Insulin (ng/mL)	6.6 ± 0.5	4.2 ± 0.5 **	2.4 ± 0.3 ***
Free fatty acid (µEq/L)	610.4 ± 78.8	450.8 ± 33.8	493.4 ± 26.2
Total cholesterol (mg/L)	780 ± 27	655 ± 28 **	725 ± 15
Adiponectin (µg/L)	27 ± 3	42 ± 4	53 ± 4 **
TNF-α (pg/mL)	178.5 ± 22.6	137.1 ± 15.1	63.5 ± 8.3 *
Resistin (ng/mL)	187.6 ± 15.9	175.9 ± 15.9	111.4 ± 11.0 **
Leptin (ng/mL)	6.8 ± 0.4	5.9 ± 0.7	6.7 ± 0.8

Each value represents the mean ± S.E. (*n* = 8). Significantly different from HFD-control: * *p* < 0.05, ** *p* < 0.01, *** *p* < 0.001 (Tukey’s HSD). HFD: high-fat diet TNF-α: tumor necrosis factor α.

**Table 4 molecules-28-01964-t004:** Gene expression analysis by real-time PCR in adipose tissue after a three-month administration of ELE in HFD-fed rats.

	Fold Change to Control	
**Gene name (Accession No)**	**3% HFD-ELE**	**9% HFD-ELE**
Perirenal white adipose tissue		
PPAR peroxisome proliferator-activated receptor (NM013124)	1.14 ± 0.13	1.43 ± 0.11 *
Adiponectin (NM144744)	1.41 ± 0.12 *	2.20 ± 0.23 *
Brown adipose tissue		
UCP1(NM019354)	1.01 ± 0.12	1.49 ± 0.18 *
UCP2(NM013167)	0.88 ± 0.43	1.13 ± 0.31

Each value represents the mean ± S.E. (*n* = 6). Significantly different from Control: * *p* < 0.05 (Tukey’s HSD). PCR: polymerase chain reaction, PPARγ: peroxisome proliferator-activated receptor γ. UCP: uncoupling protein.

**Table 5 molecules-28-01964-t005:** Effects of ASP on physical and plasma parameters after a three-month administration in HFD-fed rats.

		Diet (HFD)		
	Control	0.03% ASP	0.1% ASP	0.3% ASP
Physical parameters				
Initial body weight (g/rat)	71.0 ± 1.0	71.2 ± 1.5	72.5 ± 0.5	71.0 ± 0.6
Food intake (g/day/rat)	27.8 ± 2.2	21.3 ± 3.2 *	17.7 ± 2.7 *	14.9 ± 2.0 *
Final body weight (g/rat)	564 ± 9	516 ± 19 *	465 ± 8 *	461 ± 7 *
Body weight gain (g/rat)	493 ± 10	445 ± 18 *	393 ± 8 *	390 ± 7 *
Relative WAT weight (%)				
Perirenal white adipose tissue	2.7 ± 0.3	1.5 ± 0.2 *	1.4 ± 0.1 *	1.3 ± 0.1 *
Epididymal white adipose tissue	2.6 ± 0.2	2.5 ± 0.2	2.2 ± 0.1	2.0 ± 0.1
Plasma parameters				
Glucose (mg/L)	1621 ± 71	1501 ± 37 *	1394 ± 42 *	1338 ± 55 *
Insulin (ng/mL)	7.7 ± 0.6	5.2 ± 1.1 *	3.9 ± 0.8 *	3.3 ± 0.6 *
Free fatty acid (µEq/L)	639.1 ± 33.7	449 ± 56.0 *	402.7 ± 21.6 *	397.3 ± 20.9 *
Total cholesterol (mg/L)	880 ± 34	721 ± 25 *	708 ± 24 *	664 ± 26 *
Adiponectin (µg/L)	29 ± 5	39 ± 6 *	48 ± 4 *	53 ± 3 *
TNF-α (pg/mL)	198.3 ± 18.2	136.5 ± 13.1 *	98.7 ± 9.2 *	70.6 ± 8.9 *

Each value represents the mean ± S.E. (*n* = *6*). Significantly different from HFD-control: * *p* < 0.05 (Tukey’s HSD).

**Table 6 molecules-28-01964-t006:** Gene expression analysis by real-time PCR in adipose tissue after a three-month administration of ASP in HFD-fed rats.

	Fold Change to Control		
**Gene name (Accession No)**	**0.03% HFD-ASP**	**0.1% HFD-ASP**	**0.3% HFD-ASP**
Perirenal white adipose tissue			
PPARγ (NM013124)	1.18 ± 0.02	1.80 ± 0.12 *	2.11 ± 0.08 *
Adiponectin (NM144744)	1.01 ± 0.02	2.30 ± 0.22 *	3.03 ± 0.14 *
Brown adipose tissue			
UCP1 (NM012682)	0.98 ± 0.04	1.38 ± 0.10	1.98 ± 0.07 *
UCP2(NM019354)	1.10 ± 0.07	1.42 ± 0.13 *	1.84 ± 0.07 *
UCP3(NM013167)	0.96 ± 0.06	0.92 ± 0.06	0.89 ± 0.04

Each value represents the mean ± S.E. (*n* = 6). Significantly different from control: * *p* < 0.05.

## Data Availability

Copyright permission was obtained for the use of the figures in this review from the Life Science Publishing Co., Ltd.

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
