# Peer review of "The Differences of Mechanisms in Antihypertensive and Anti-Obesity Effects of Eucommia Leaf Extract between Rodents and Humans"

_molecules, 2023, doi:10.3390/molecules28041964_

Round 1

Reviewer 1 Report

The manuscript analysis the effects of geniposidic acid, a major iridoid component of Eucommia leaf extract on certain pathophysiological mechanisms. The general appearance of the works is superficial, done in haste, without attention to details. Both content and presentation always matter. The expression must be carefully revised and corrected, it is confusing many times, the authors using terminology improper for a review, but for an article, even they declared their paper as being a Review. Some figures are simplistic. The tables need to be better thought out and completed. Some of the figures/tables can be merged, to be more complex. Please see below my suggestions regarding this manuscript, which requires a more than extensive revision:

1.              L49-53. The authors have described what they have done in the study, but they have not presented the aim of the study, novelty/special contributions it brings to the field, the reason for choosing this topic. Please provide the aim of the study. The last paragraph of the Introduction section must contain precisely defined the purpose of the work.

2.              2nd section begins with Figure 1. In a scientific paper BEFORE inserting a Figure/Table, it must be mentioned in the main text. Please proceed consequently. There are more figures/tables in the manuscript requiring to be introduced first in the text.

3.              Table 1. This being a Review type manuscript, it requires a last column of Ref. (references). Moreover, for 3 numerical values, it is not necessary a Table. You have 2 options: a. improving and providing a strong information table, or b. removing the table and inserting a simple sentence with those 3 numerical values of the compounds. Moreover, for all Tables you will present, it is requested a last column of references. (this is a Review; each statement must be supported by the proper reference)

4.              Figures 4-7, 9.  Acronyms/Abbreviations/Initialisms should be defined the first time they appear in each of three sections: the abstract; the main text; under the first figure or table. When defined for the first time, the acronym/abbreviation/initialism should be added in parentheses after the written-out form. Please revise and check all the Abbreviations to be explained at their first mention in the text, including UNDER the mentioned above figure, after the title. Please check and revise the entire manuscript in this regard.

5.              Tables 6 and 7. There is no head of the table for the 1st column. Please provide.

6.              The other sections can be improved. Please discuss the extract as an antidiabetic agent given the fact that it increases GLP-1 level. I suggest checking and referring to PMID: 32765722  Also, please detail the potential role of the plants (as an atherosclerosis preventive agent), maybe in the novel formulation (nano), given the fact that it reduces blood pressure and has many beneficial effects in the pathogenesis of this process; in this regard I suggest PMID: 35687909 ; https://doi.org/10.3390/chemosensors9040067

7.              L427. You can not postulate anything. A review presents data and conclusions of already published papers. To postulate means that is about your original research.

8.              Limitations do not need a separate section. 

9.              L477. Moreover, you have not demonstrated nothing. The presented results are not yours. You may conclude but not demonstrated 

10.            L483. You have not found.

11.            L485. You have not reported nothing for the first time, in this manuscript, as this is a Review. Others did report the data you have summarized/presented/depicted/described.

12.            L491. You have not demonstrated nothing. There are not your original data/info.

Author Response

Responses to comments 

We thank you for the kind comments.

The comments pointed out have been corrected as follows.

The corrected parts are written in red for each applicable page in the manuscript.

Reviewer (1)

This review is mostly summarized from our new findings based on our original papers which are listed in the References.

 The first original manuscript has already been checked and proofread by native English-speaking staffs in professional Egona.

 The part of text in this manuscript, written in black letters has already been checked and proofread in the original manuscript by professional Egona. The parts of text written in red in this manuscript are the parts of text which were newly revised this time..

  1. L-49-53 in the original manuscript

By the suggestions from the Reviewer, this part of text was rewritten. In addition, we added our purpose and conclusion of this review into Introduction and Conclusion, respectively.

  1. Some of Figures and Tables have been rearranged in this manuscript as pointed out by the Reviewer..

Most of the figures are reprinted from the original papers, and we have received permission from the publisher to use them, so we would like to use them as they are.

  1. In order to easily understand the effects ,the data written in each of Tables are serected from the original papers as the important data which support the action of antihypertensive effect of ELE in endothelium and the action of anti-obesity in WAT and BAT, respectively. The similar styles are performed in our previous Molecules paper (Molecules 2021, 26, 2362). So we would like to use them as they are.

Table 1 in the original manuscript was omitted from this manuscript and only PDG data was added in the text.

  1. 4. The revision to the formal name (acronym) in the text was rewritten in red letter as pointed out by the Reviewer.

  1. The heads of the table for the 1st column in Table 3 and Table 5 in this manuscript were added, respectively.

  1. This Review focuses only on the antihypertensive and anti-obesity effects and the these mechanisms of ELE. This time, we have not reviewed ELE as an antidiabetic agent or as an atheroslerosis preventive agent, We have obtained data on the potential of these agents. The review as these agents is considered necessary in the future.

  1. L-427 in the original manuscript. In order to compare the action of GEA with that of infliximab in the humans clinical traial, here we described the data of infliximab reported in the paper. Actually, our research collaborator, Hosoo et al performed the humans clinical trial and compared our data of GEA with data of inflaximab pablished in paper.

The reason for comparing the data of GEA and infliximab was added in the text based on the original papers. 

These our original papers are listed in the Reference.

Since there are no reports of human clinical trials regarding on Torenia extract, the part  L-448-450 of the text in this manuscript has been rewritten to a simple description.

Figure 9 in the first manuscript was omitted from this manuscript.

 Citation numbers 47-53 of Reference in the original manuscript have been changed to the new citation numbers 47-53 of Reference in this manuscript due to the rewriting.

  1. “Limitations” in the original manuscript was omitted from this manuscript as pointed out by the Reviewer.

  1. As the a part of the mechanism, we had found and reported that asperuloside elevates bile acid pools in the ileum. This is due to the inhibition of bile acid transporters by asperuloside hydrolysates in the ileum. So we don't think these findings mean that we haven't demonstrated anything.

  1. . We will send the copy of our original paper to the Reviewer under the separate e-mail, if needed.

11.L-485. Our research collaborators carried out the humans clinical traials in Chinese subjects with Chinese researchers in China. The part of L-481-485 text in this manuccript were rewritten as such. These data are not published by other researchers other than our researchers as the original paper listed in the References

  1. L-491. We will send the copy of our original paper to the Reviewer under the separate e-mail, if needed.

 We wish that this revised manuscript will be accepted in Molecules for the publication.

  Prof. D. Sansei Nishibe

Reviewer 2 Report

Comments and suggestions for Authors

The review entitled “The differences of mechanisms in antihypertensive and anti-obesity effects of Eucommia leaf extract between rodents and humans” authored by Sansei Nishibe et al focuses on the mechanisms behind the anti-hypertensive and anti-obesity benefits of Eucommia leaf extract and lists the variations in these mechanisms between effects on rats and humans based on the so far research.

The review study is very interesting since it presents data that may be helpful in understanding how specific molecules from Eucommia leaf extract, that may exist in other edible plants, may mediate mechanisms concerning obesity and hypertension that are significant public health challenges worldwide.

The review is well organized and describes well and in a comprehensive way the data from the so far research  

I have a minor comment before it can be accepted for publication:

Minor comments

1.          Authors should check if 150% is the correct level (lines 91-92)

Author Response

Responses to comments   

We thank you for the kind comments.

The comments pointed out have been corrected as follows.

The corrected parts are written in red for each applicable page in the manuscript.

Reviewer (2)

We thank you for your high evaluation of our Review.

We corrected the point pointed out by the Reviewer (L-101; 150% → 50%).

We wish that this revised manuscript will be accepted in Molecules for the publication.

 Prof. Dr. Sansei Nishibe

Round 2

Reviewer 1 Report

The authors revised parts of the paper based on the suggestions received, but there are still parts where they did not make changes but rather tried to provide explanations in the letter to the reviewer. Please revise it more comprehensively based on the suggestions.

After rechecking the authors' response, I believe that the initial suggestions have been fulfilled and have contributed to the quality of the manuscript, making it suitable for publication. However, self citations must be reduced. Much too many.

Author Response

We thank you for the kind comments.

We have been researching the antihypertensive and anti-obesity effects of Eucommia leaf extract and their active components for more than 20 years.

During this process, we discovered for the first time that the antihypertensive effects in rodents and humans were due to geniposidic acid, but the mechanisms of action were different. To date, no one other than us has reported the papers on humans clinical trials of Eucommia leaf extract for the antihypetentive and anti-obesity effects. Also there are no reports on ANP secretion due to geniposidic acid in rodents and on their mechanisms. Similarly, there are no reports other than our research reports on anti-obesity effects.    This Review summarizes our research of Eucommia leaf extract over more than 20 years, and inevitably contains many self-citations in the Review.

We are also researching the active components in Eucommia bark. Here, we have deleted the description of the the bark-specific component PDG as cAMP-PDE inhibitor and our Reference 16 from the manuscript. Further we are doing research on Plantago lanceolata. We also wrote about the catalpol of Plantago in this Review.. The description of this part of Plantago and References were also deleted from the manuscript, since this Review is focused on our research of Eucommia leaf extract as Eucommia leaf tea developed from roasted Eucommia leaves..Therefore, there are almost no unnecessary self-citations. So far, there are no similar papers other than our papers of Eucommia leaf extract. All our original papers (peer-reviewed) which we cite in this Review are the only ones that have been done the experiments and reported by our research collaborators. 

We deleted the description of the parts of Eucommia bark and of Plantago(L-150-163 of the original manuscript)and removed Refs 16, 18 and 19 from the original manuscript..

 The reference numbers have been partially changed due to the revision, so, they are written in red in the revised manuscript.

We wish that this revised manuscript can be accepted in Molecules for the publication.